# Simulation and prediction of rural population changes using agent-based modeling

**Shanshan Huang[1,2], Yao Huang[1,2], Shitai Bao [1,2]\*, Jianfang Wang[1,2], Siying Chen[3]**

1 College of Resources and Environment, South China Agricultural University, Guangzhou, China,
2 Guangdong Province Key Laboratory for Agricultural Resources Utilization, Guangzhou, China,
3 College of Forestry and Landscape Architecture, South China Agricultural University, Guangzhou, China

\* bst100@scau.edu.cn

## Abstract

Rural population change is a critical element of the strategy for rural revitalization in China. Many studies emphasize large-scale macro-population trends, but a noticeable gap exists in micro-level simulations and predictions regarding rural population size and structure. This study employs an agent-based model(ABM), defining a population agent and its behavioral rules. By modeling individual-level birth, death, and migration behaviors, it generates agent-based outputs that aggregate to capture population dynamics and forecast rural demographic trends over the next 11 years. Using two representative villages as study areas, the results were validated by comparing them with actual population data and predictions made by the Leslie model. The findings demonstrate the following: 1) the agent-based modeling effectively captures the dynamics of births, deaths, and migrations at the micro level, elucidating the underlying determinants of rural population retention. 2) In economically disadvantaged villages, the total population, labor force, and proportion of adolescents have significantly declined. Notably, emigration is pronounced in villages without industrial advantages, regardless of substantial per capita arable land; the youth labor force constitutes less than 30%, while the aging population is as high as 45%. 3) Migration and birth rates are key factors influencing rural population trends. To mitigate future rural population aging, enhancing birth rates and fostering rural industrial development is essential to curb migration. These findings support evidence-based policies to stimulate birth rates, attract and retain younger populations, and enhance economic opportunities in rural areas. The micro-level analysis enables the design of more effective and context-specific rural revitalization programs, bridging the gap between micro-level behaviors and macro-level demographic patterns.

**Data availability statement:** The datasets generated and analyzed during this study contain sensitive participant information, including age and gender, which are protected under institutional ethical guidelines and data privacy regulations. Due to these legal/ethical restrictions, raw data cannot be made publicly available. However, de-identified data supporting the findings are available upon request to qualified researchers through contacting the corresponding author or the Academic Board of the College of Resources and Environment, South China Agricultural University (zhxyky@scau.edu.cn) with proof of institutional review board approval and a signed data use agreement.

**Funding:** The author(s) received no specific funding for this work.

**Competing interests:** NO authors have competing interests Enter: The authors have declared that no competing interests exist.

## Introduction

Since China's reform and opening up, its rural population has increased from 790 million in 1978–860 million in 1995, rapidly declining to 510 million by 2020. This decline and accelerated urbanization resulted in substantial migration from rural areas to cities. Understanding population dynamics is essential for the sustainable development of rural areas [1]. Studying rural population dynamics requires the analysis of historical data and predictive modeling of trends in population change [2].

Studies have thoroughly examined the total rural population and its structure from various cultural and demographic perspectives across different scales [3]. Meanwhile, existing research has predominantly focused on rural population flow and urban agglomerations [4–5], while changes in the population at a more localized rural scale have received less attention. Most analyses have employed large-scale administrative regions to cover long time series for calculating distributions and changes in urban and rural populations. For instance, Jiang hui and Shun studied the Chang-Zhu-Tan urban agglomeration based on total population data and current distribution, extrapolating trends, and the scale of rural population migration in Hunan Province from 2016 to 2030 [6]. Other studies have centered on forecasts and autocorrelation indicators related to population size and structure [7]. While these research indicators of population change are varied, they often display significant homogenization.

The analytical methods used to study population change primarily rely on macro statistical data and mathematical models. Common techniques include time-series analysis [8], regression analysis [9], and various demographic models, such as the Malthus population index model [10], the logistic model [11], the Leslie model [12], the ARIMA prediction model [13], and the cohort component method [14–17]. Xinyi and Haojuan (2017) investigated population characteristics using neural networks to forecast total population, birth rates, death rates, and the elderly; however, the models lacked sufficient sensitivity to certain macro-level population factors, leading to significant prediction errors [18]. Each method has distinct advantages and disadvantages. For instance, the Malthus population index model effectively predicts exponential growth or decline based on the birth-to-death ratio but overlooks the influence of various factors, such as economic conditions, on population dynamics. The logistic model accounts for limitations in population growth and examines patterns of change; however, it tends to produce large and unstable errors over extended prediction periods. The Leslie model considers the internal population structure across different developmental stages, yielding reasonably accurate results in various scenarios. Yet, it struggles to capture nonlinear changes in fertility and mortality rates at specific ages. Additionally, the ARIMA model has significant limitations, as it can only model and predict a single population factor at a time, ignoring the impact of other population-related factors on future size and structure [19,20].

These models possess specific advantages rooted in particular theoretical frameworks and application contexts; they also have the following limitations in addressing complexities inherent in population dynamics: (1) Traditional prediction models often assume static and linear population behavior, which hampers their ability to

accurately represent the dynamic changes in the population structure; (2) These models concentrate on macro-level predictions, failing to capture internal differences at the micro level, which complicates the simulation of nuanced population changes under specific policies or environmental conditions.

In recent years, simulation approaches based on individual-level behaviors have offered new perspectives for addressing the aforementioned limitations. Among these methods, the agent-based modeling effectively supports an in-depth analysis of how individual behaviors influence population dynamics. By simulating individuals' life cycles, mortality, and migration behaviors, researchers gain a deeper understanding of the quantitative and structural changes within population groups. It addresses existing research's limitations and enhances practical value for predictive outcomes. One key advantage of the agent-based modeling approach lies in its ability to directly capture dynamic developments and introduce counterfactual scenarios. Unlike traditional models that rely heavily on static statistical data, ABM enables a more realistic representation of individual behaviors and their aggregate effects on population change. ABM captures real-world population processes' nonlinear dynamics, including policy adjustments' influence [21] and social networks [22].

Moreover, this approach enhances the analytical capacity for studying rural migration patterns, enabling the simulation of individual migration decisions under varying socio-economic conditions [23]. Additionally, ABM employs time-step simulation to dynamically track life cycle events, thereby revealing the underlying drivers of population change, such as shifts in family structure and intergenerational migration patterns [24]. The individual-based nature of ABM offers the flexibility to adapt to different regional contexts by adjusting behavioral rules, ultimately improving the accuracy of regional population forecasts. Furthermore, ABM facilitates the development of diverse policy scenarios, such as agricultural decision-making frameworks [25], which allows for assessing their impacts on rural population mobility and demographic transitions.

Current bottom-up approaches to population research remain limited in scope [26]. To address the uncertainty and complexity of population change, it is essential to first gather information about common individual behaviors related to life, death, and migration, intelligently simulate changes in individual characteristics, and ultimately identify patterns of change within population groups. Accordingly, this study was conducted on a micro-scale to examine conditions such as declining birth rates, migration, and aging in rural areas. We explored the behaviors of individuals across different age groups using ABM technology, established concise and clear behavioral rules, and inferred quantitative and structural changes in the population group on a small scale through simulations of individual behaviors on a larger scale. The results were subsequently assessed using the Leslie model. We aimed to uncover the laws and trends of population change at the village scale, analyze prediction outcomes, and discuss the underlying causes driving demographic shifts in rural areas. We ultimately sought to reveal future trends in population evolution and provide critical support for planning rural revitalization.

## Methods

### Agent-based modeling

Agent-based modeling provides a sophisticated methodological framework emphasizing individual behaviors, making them particularly adept at simulating and analyzing complex systems. This approach leverages modern technology's parallel and distributed computing capabilities, allowing for precise predictions of changes in collective behavior within a population. Each agent in the model possesses distinct attributes and adheres to specific behavioral protocols, enabling autonomous decision-making based on established rules. Thus, this framework is powerful for simulating and forecasting population dynamics.

This study employed an agent-based modeling approach to investigate three fundamental behaviors in rural populations: birth, death, and migration. Population agents were characterized by essential attributes such as age, gender, and marital status. Agents from different age cohorts exhibited varied behaviors. For instance, adolescents may migrate for educational purposes, individuals of marriageable age may marry, women may bear children, adults may participate in the

workforce, and the elderly may face mortality, including instances of accidental death. Fig 1 illustrates the shifts in population structure resulting from these individual agent behaviors, effectively simulating population growth through births, decline due to deaths, and changes caused by migration. This method provided a comprehensive framework for analyzing fluctuations in agent counts and variations in age and gender distribution throughout the simulation, shedding light on underlying patterns and trends in population dynamics.

1. **Birth**: The birth of female agents contributes to the overall population growth based on their fertility periods and birth rates. As women enter their reproductive years, they encounter specific probabilities for childbirth, with a fertility rule applied for simulation purposes that allows for repeated births in subsequent steps. Furthermore, this paper defines reproductive behavior as a probabilistic event, capturing the likelihood of a woman's agent conceiving a newborn agent, with the probability of occurrence equal to that year's population birth rate, as detailed in Table 1.

2. **Death**: The population decreases due to natural mortality and other lethal factors. The number of agents is primarily determined by life expectancy and mortality rates at various ages. In this model, agents have a maximum life expectancy of 100 years; over this age, they cease to exist. Death is a probabilistic event for population agents, with the same probability value as the population mortality rate for that year, as indicated in Table 1.

3. **Migration**: This study identifies three primary categories of outward migration from rural populations: youth pursuing higher education, individuals marrying at appropriate ages, and the labor force seeking employment. Migration behavior as a probabilistic event is defined in this paper as the probability of an agent occurring in these three types of migration behaviors, and the probability settings are shown in Table 1.

### Simulation rules

To enhance understanding of rural population dynamics, this section details the definitions of the established agents' behavioral rules alongside the corresponding simulation probabilities and their underlying bases, as outlined in Table 1. Migration scenarios considered individuals aged 3–22 years pursuing education, those aged 20–40 years migrating for marriage, and individuals aged 18–60 years leaving for labor opportunities. For females aged 22–40 years, the simulated birth probability was set at 6.77‰, consistent with the national birth rate in rural areas. Additionally, the probability of

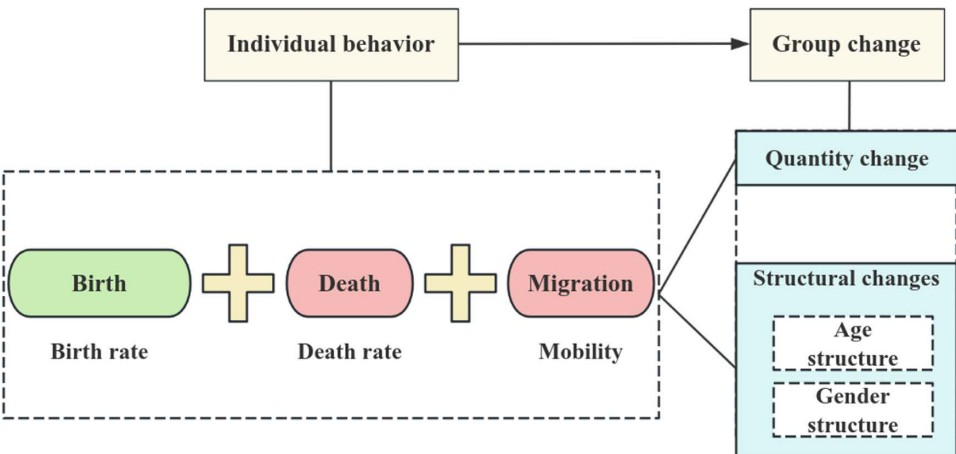

**Fig 1. Individual population behavior and group changes using agent-based simulation.**

**Table 1. Behavioral rules and probability of population agents.**

| Age group | Behavioral rules | Probability of population migration | Probability values | Source reference for those probabilities |
|---|---|---|---|---|
| Ages 3–22 | Migration for further education | Migration rate for primary and secondary schools | 4.66–9.76% | Statistical Bulletin on the Development of Education in China in 2022 |
| | | Enrollment rate of college students | 0–48.6% | |
| Ages 20–40 | Marriage migration | National/local marriage rate and proportion of marriage registrations | 5.4–9.9‰ | Statistical Bulletin on the Development of Civil Affairs in 2023 |
| Ages 19–59 | Rural people are migrating to cities for work | Migration rate = 1 − Ratio of permanent resident population | 0–25% | Guangdong Statistical Yearbook (III. Population 3–6 Changes in Household Population Migration) |
| Ages 22–40 | Birth rules for married women of marriageable age | Average birth rate of the population | 5.73–9.35‰ | National Bureau of Statistics |
| Ages 0–100 | Death rate of all agents | Accidental mortality of young and middle-aged | 0.55–0.95‰ | China Health Statistical Yearbook 2022 |
| | | Mortality rate of the elderly | 4.83–8.7‰ | |

individual extinction was aligned with national mortality rates, which were set at 0.55‰ for young and middle-aged individuals and 8.5‰ for the elderly.

## Population prediction models

The Leslie model is a comprehensive framework for predicting the population sizes and age structures of one or more regions based on initial population data categorized by age and gender. This model is typically used for predicting and validating population age groups. For the female population, let $x_i(t)$ represents the total number of females in age group $i$ during the $t$ observation period. The fertility rate for females in age group $i$ is denoted as $b_i$, and the female mortality rate is represented by $d_i$. The analysis assumes villages as the units of study, and time variations in the validation of $b_i$ and $d_i$ are calculated in conjunction with actual different age groups. The temporal changes in the validation process are then calculated and integrated with the actual age groups. Additionally, these time changes are verified using real age-specific population out-migration factors, ensuring that the model accurately reflects demographic shifts influenced by migration patterns. This approach enhances the reliability of the population predictions by incorporating observed data on out-migration specific to different age cohorts. The prediction formula for village populations across various age groups is expressed as follows,

$$\begin{cases} x_{0j}(t+1) = \sum_{i=0}^{n} b_{ij}(t)w_{ij}(t)x_{ij}(t) \\ x_{ij}(t+1) = x_{ij}(t)[1 + p_{ij}(t) - m_{ij}(t)] \end{cases} \tag{1}$$

where $j$ denotes a village, $t$ represents the year, and $i$ indicates a specific age group of the population. The population is classified into n groups, from youngest to oldest (0–100). $x_{0j}(t+1)$ represents the birth number in village $j$ in the $(t+1)$ year, which is mainly determined by the fertility rates of women in each age group in year $t$ and their age structures; $x_{ij}(t)$ is the quantity of the population in age group $i$ in village $j$ in year $t$; $b_{ij}(t)$ is the annual fertility rate of women in age group $i$ in village $j$ in year $t$ ($b_{ij}(t)$ = the number of infants born in age group $i$ in year $t$/the number of women in age group $i$ in year $t$); $i$ = 1, 2,..., n (that is, the proportion of women giving birth in age group $i$ in year $t$); $w_{ij}(t)$ is the proportion of women in age group $i$ in village $j$ in year $t$, and $x_{ij}(t+1)$ is the quantity of the population in age group $i$ in the $(t+1)$ year, influenced by factors such as fertility, mortality, and migration; $p_{ij}(t)$ is the survival rate of the population in age group $i$ in village $j$ in year $t$ ($p_{ij}(t)$ = 1 is the number of deaths in age group $i$ in year $t$/the number of people in age group $i$ in year $t$); $m_{ij}(t)$ is the migration rate of the population in age group $i$ in village $j$ in year $t$ ($m_{ij}(t)$ = the number of emigrants in age group $i$ in year $t$/the number of people in age group $i$ in year $t$), that is the proportion representing emigration.

## Population simulation

### Overview of the study area

The study area, Xiache Town in Heping County, Guangdong Province, is located on the outer fringes of the Pearl River Delta and is categorized as a relatively underdeveloped region. Geographically adjacent to Jiangxi Province, Xiache Town encompasses an area of 14,310 ha and is predominantly characterized by hilly topography. The following provides a fundamental overview of the two villages in the study area.

Yunfeng Village is located in the western part of the town and covers an area of 1527 ha. It comprises three village groups and has current resident population of 1060. The total area of arable land is 93.93 ha, while forested land totals 1,245.3 ha, contributing to its classification as a typical mountainous village. In recent years, Yunfeng Village has experienced significant development in the kiwifruit-based industry, with a kiwifruit cultivation area of 720 ha out of a total fruit cultivation area of 853.33 ha.

Shihan Village is located in the northeastern part of the town, covering an area of approximately 1700 ha, comprising seven village groups and current resident population of 1319. Due to limited industry in this village, most residents must seek employment elsewhere. The research team gathered the registered population and resident population of the two villages in 2017 and in 2024. Data was collected through multiple channels, including township police stations, village committees, and natural resource management departments. The grid-based management household information card recorded detailed information on the age, gender, and emigration destinations of each household's resident population, providing an opportunity for thorough population simulation and validation. The basic situation of the two villages is displayed in Table 2, and the resident population structure is illustrated in Fig 2. Because of the unique household registration system in rural China, the actual resident population living in the countryside is much smaller and more real than the household registration population. This paper therefore only simulates and predicts the resident population.

The population pyramids for 2024 revealed distinct demographic profiles for both villages. The population pyramid of Yunfeng Village has a wider base, indicating a higher proportion of youths aged 0–14 years, a significant number of middle-aged individuals aged 45–54 years, and a smaller elderly population aged 65 and above; this suggests that Yunfeng Village has a younger population structure with a larger workforce of middle-aged individuals. Conversely, the population pyramid of Shihan Village exhibits a narrower base, indicating a smaller proportion of people aged 0–14 years, fewer individuals in the working-age group of 40, and more elderly residents over 60.

### Simulation process

To simulate the village's population dynamics, we utilized the GAMA platform and integrated ABM technology, as shown in Fig 3. The 2017 demographic data served as the baseline for model operations, defining agents for the entire village population and assigning them appropriate age and gender attributes. We simulated three primary behaviors- birth, death, and migration based on age and gender, with one year representing the simulation time step. This approach yielded the predicted village population for 2024. We subsequently verified the results against actual population data and predictions generated by

**Table 2. Basic information of the two villages in the research area.**

| | Resident population/ (persons) | Proportion of young population (aged ≤ 14 years) | Proportion of young and middle-aged population (aged 15–64 years) | Proportion of senior citizens (aged ≥ 65 years) | Per capita annual income (¥) | Total area of cultivated land (ha) | Per capita cultivated area (m²/person) |
|---|---|---|---|---|---|---|---|
| **Yunfeng Village** | 1060 | 38.49% | 52.26% | 9.25% | 72000 | 93.93 | 886.17 |
| **Shihan Village** | 1319 | 12.21% | 63.23% | 24.56% | 12140 | 153.73 | 1165.54 |

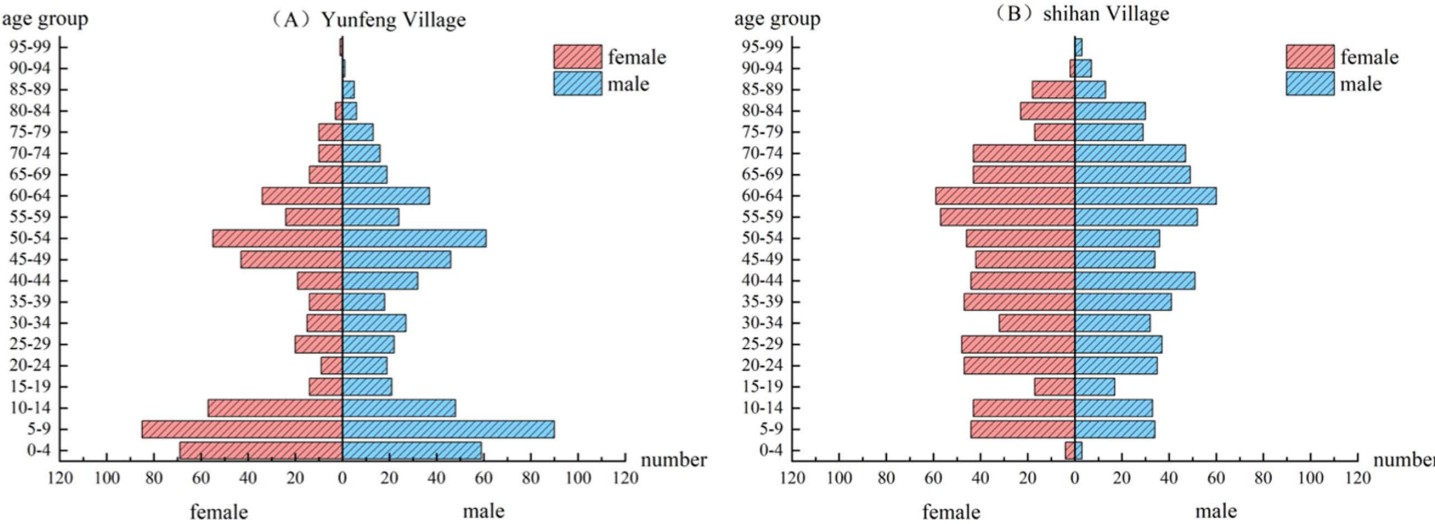

**Fig 2. Resident population pyramid of the two villages.**

**Fig 3. Process of simulating and predicting the rural population changes using agent-based modeling.**

the Leslie model. This simulation also enabled predictions of the village population from 2025 to 2035, analyzing total population trends, gender distribution, and age structure to explore the driving factors behind population changes.

The GAMA platform is a comprehensive tool that integrates spatial, multiparadigm, and multiscale simulations while supporting large-scale simulations and processing thousands of agents in parallel [27]. Consequently, it was selected to define population agents and conduct simulations and predictions for village populations. The operational interface is shown in Fig.4.

**Verification of simulation results**

We undertook a systematic verification process to validate the accuracy of the village population simulation results. First, we compared the simulation results with current demographic data. Subsequently, we conducted a comparison between the simulation outputs of the Leslie model. Tables 3 and 4 present the prediction and simulation results of the two models and the findings. The two models for the total population are aligned with prevailing population statistics. The Leslie model forecasts the demographic structure of the population based on the population gender ratio, fertility rate, and mortality rate. The results of the agent-based model (ABM) simulation were closer to the current resident population of all age groups than the results of the Leslie model.

The paired sample T-test was conducted on the difference between the simulated values and the resident population in each age group across the two villages using SPSS 27 software. The results of the paired samples T-test presented in Table 5 suggest that the null hypothesis, which states that the simulated values are equal to the actual values of the resident population, should be accepted at the 95% confidence level. The robustness of the paired samples T-test was assessed in Table 6 using bootstrapping, a method for deriving robust estimates of standard errors, and confidence

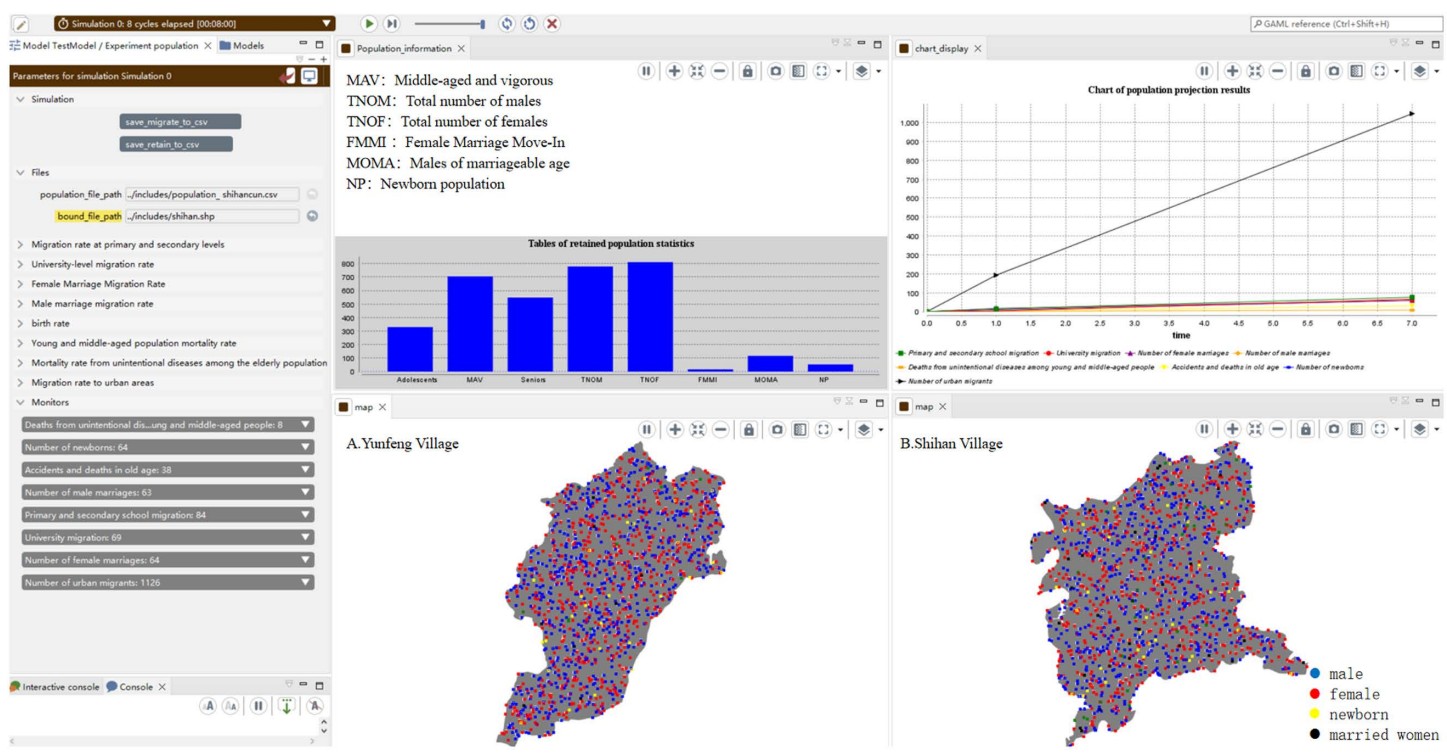

**Fig 4. Operational interfaces in GAMA platform for rural population simulation.**

**Table 3. Comparison of simulation result verification in Yunfeng Village in 2024.**

| Age group (years) | ABM simulation/(persons) | Leslie model/(persons) | Current resident population/(persons) |
| --- | --- | --- | --- |
| Age ≤ 19 | 379 | 102 | 443 |
| 20–30 | 98 | 238 | 73 |
| 31–40 | 102 | 171 | 75 |
| 41–59 | 302 | 324 | 300 |
| Age ≥ 60 | 180 | 180 | 169 |
| Total population | 1061 | 1015 | 1060 |

**Table 4. Comparison of simulation result verification in Shihan Village in 2024.**

| Age group (years) | ABM simulation/(persons) | Leslie model/(persons) | Resident population/(persons) |
| --- | --- | --- | --- |
| Age ≤ 19 | 183 | 180 | 195 |
| 20–30 | 175 | 210 | 178 |
| 31–40 | 112 | 200 | 159 |
| 41–59 | 331 | 314 | 344 |
| Age ≥ 60 | 523 | 450 | 443 |
| Total population | 1324 | 1355 | 1319 |

**Table 5. Paired samples T-test.**

| | Paired Differences | | | | | t | df | Sig. (2-tailed) |
| --- | --- | --- | --- | --- | --- | --- | --- | --- |
| | Mean | Std. Deviation | Std. Error Mean | 95% Confidence Interval of the Difference | | | | |
| | | | | Lower | Upper | | | |
| Pair 1 simu – acuate | 1.000 | 36.332 | 10.488 | −22.084 | 24.084 | .095 | 11 | .926 |

**Table 6. Bootstrap check for paired samples T-test.**

| | Mean | Bootstrap[a] | | | | | |
| --- | --- | --- | --- | --- | --- | --- | --- |
| | | Bias | Std. Error | Sig. (2-tailed) | 95% Confidence Interval | | |
| | | | | | Lower | Upper | |
| Pair 1simu - acuate | 1.000 | .268 | 10.209 | .929 | −19.746 | 22.831 | |

[a]. Unless otherwise noted, bootstrap results are based on 1000 bootstrap samples

intervals for estimates such as the mean, median, proportion, correlation coefficient, or regression coefficient. Additionally, the bootstrap results based on 1,000 samples indicate that the robustness check was successfully passed.

To explore the gender structure of the population, we compared the ratios of males and females in the simulation results with those of the resident population. The results in Fig. 5 demonstrated that the simulated value of the gender ratio of the two villages in 2024 is basically consistent with the actual situation, and there is little difference between them. This observation confirms the simulation's accuracy in reflecting gender distribution.

## Results and discussion

### Comparison of simulation results

With consistent birth rate of 7.66‰ and a death rate of 8.5‰, the populations of the two villages were simulated from 2018 to 2023, and the trends in the proportions of adolescents, young and middle-aged adults, and elderly individuals were analyzed, as illustrated in Fig.6.

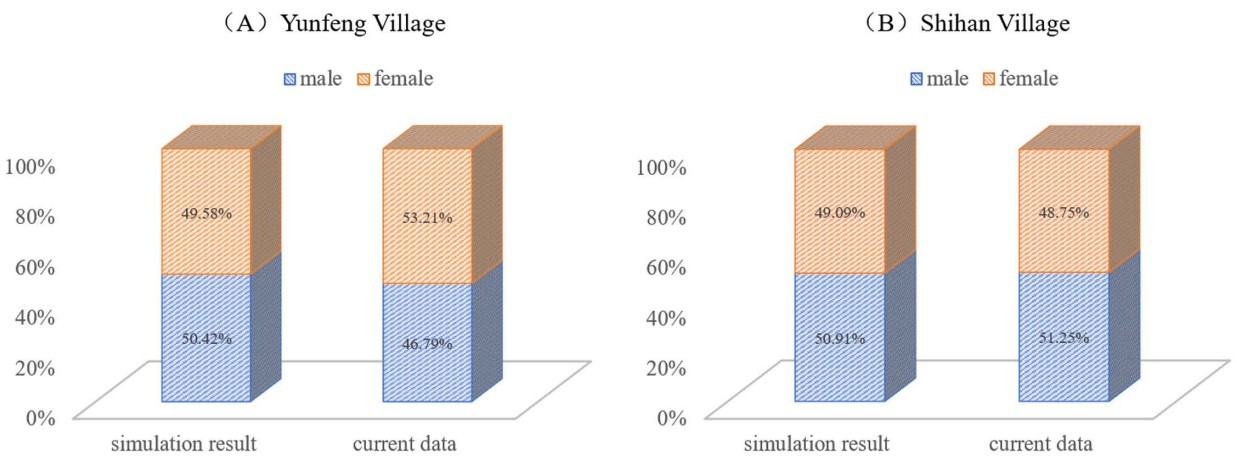

**Fig 5. Comparison of simulation verification of gender structure of rural population.**

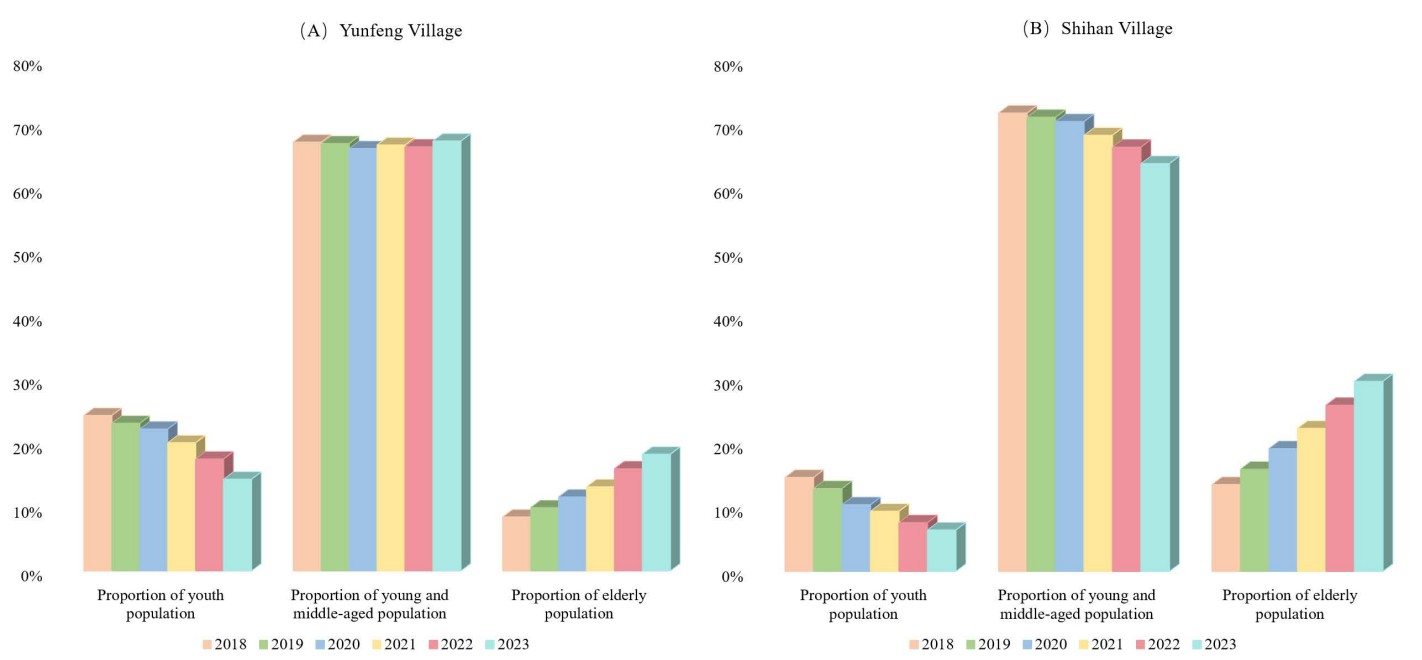

**Fig 6. Results of population structure changes in two village types.**

The analysis revealed a steady decline in the proportion of adolescents in both villages over the observation period, which is largely attributed to persistently low birth rates in these areas. Notably, Yunfeng Village, benefiting from industrial development, exhibited a higher proportion of adolescents than Shihan Village, which lacks industrial infrastructure. Conversely, the proportion of elderly individuals increased in both villages, indicating a growing aging trend, particularly in Shihan Village, where the elderly population surpassed 20% and continued to rise rapidly; this highlights the more severe aging issue in villages lacking industrial support. The proportion of young and middle-aged individuals in Shihan Village declined annually due to out-migration. In contrast, Yunfeng Village maintained a stable proportion of young and

middle-aged individuals over the five years, attributed to robust local industries and ample employment opportunities. The kiwifruit cooperative in Yunfeng Village exhibits a high degree of specialization, yielding greater economic benefits than external employment. Consequently, all households participated in kiwifruit cultivation, resulting in minimal population outflow.

## Prediction of population structure changes

Based on historical simulation methods and demographic parameters of Yunfeng and Shihan villages, and assuming birth and death rates of 7.66‰ and 8.5‰, respectively, the populations of the two villages were predicted accurately from 2025 to 2035. The predicted proportional changes in teenagers, middle-aged individuals, and the elderly are depicted in Fig 7. Population decline, labor force outflow, and demographic aging remain prevailing trends. In terms of total population,

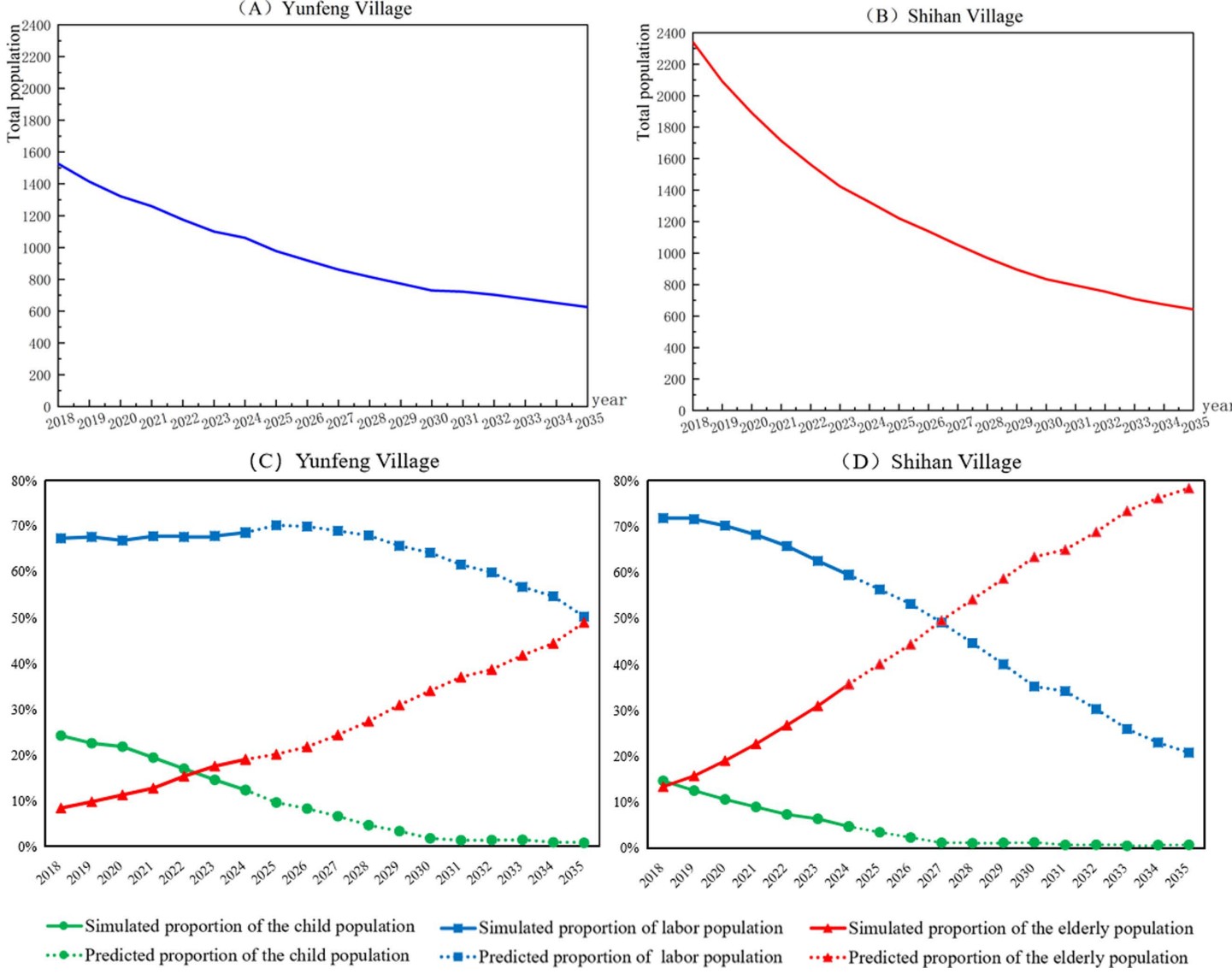

**Fig 7. Predicting the retained population of villages in the next 11 years based on agent-based simulation.**

Shihan Village exhibits a significantly faster decline rate compared to Yunfeng Village, which benefits from industrial advantages. Regarding demographic composition, the youth population in both villages has decreased, reflecting a lower birth rate in rural areas. Simultaneously, both villages show increasing elderly population ratios and evident aging trends. However, Shihan Village presents substantially higher proportions and growth rates of elderly residents than Yunfeng Village. While young and middle-aged adults are declining in both villages, Yunfeng Village experiences notably milder reduction in both magnitude and pace. This comparative analysis reveals that villages with industrial foundations can better retain working-age populations through abundant employment opportunities and higher incomes, thereby mitigating outmigration rates, helping maintain rural fertility rates. In contrast, Shihan Village, lacking industrial support, faces exacerbated demographic challenges: the exodus of young and middle-aged workforce leaves primarily elderly residents engaged in agricultural production, accelerating population aging. This shift has worsened challenges related to an aging population, including abandoned farmland and a declining rural labor force. [28].

Several trends emerge from analyzing the age and gender structure of the rural population in the two villages over the next 11 years, as shown in Fig 8. Between 2024 and 2035, the age structure of the population in Yunfeng Village remains relatively stable overall. Specifically, the proportion of individuals aged 15–19 shows a gradual decline, while the share of those aged 70 and above steadily increases. The dominant age cohort shifts from individuals aged 10–19 in 2025 to those aged 20–29 by 2035. This structural evolution indicates a relatively low rate of outmigration, attributed to the village's modest industrial base and the availability of local employment opportunities, which contribute to the stability of its resident population. Nevertheless, Yunfeng Village also faces the challenge of persistently low fertility rates. The continuous decline in the 0–4 age group throughout the forecast period suggests a potential contraction in the overall population size in the future. In terms of gender composition, the male-to-female ratio remains relatively balanced across the general population. However, among individuals aged 60 and above, females consistently outnumber males, which may place increasing demands on future elderly care services and resource allocation.

During the forecast period, Shihan Village—lacking industrial support—experiences a high rate of outmigration, leading to an increasingly inverted age structure with evident aging and youth loss. The outflow of working-age individuals weakens both fertility potential and labor supply, resulting in a regressive population structure. This structural decline is reflected not only in the reduction of total population but also in the deterioration of population vitality. In particular, the continued decline in the number of children aged 0–4 and 5–9 indicates persistently low fertility rates. By 2030, approximately 500 residents will exit the labor force while only 99 will enter, causing a severe imbalance in labor replacement. The aging rate is projected to reach 70%, with the 65–74 age group becoming the dominant cohort. These changes stem from geographic and economic constraints—fragmented, low-yield farmland in mountainous terrain, limited arable land per capita, and a lack of local employment—driving young people to migrate. As a result, the village faces a vicious cycle of depopulation, labor loss, and agricultural decline Fig 9.

**The impact of births, deaths and migration rates on population change**

**Impact of birth and death rates on rural population changes.** In the population simulation projections discussed above, the birth and death rates directly influence the evolution of the population structure. Given that medical and health conditions in China are improving, life expectancy is steadily increasing, and mortality rates remain consistently low and stable, our focus is primarily on the impact of birth rates on demographic changes in rural populations. The low birth rate has resulted in a declining proportion of adolescents and young people, a gradual labor force contraction, and a clear trend toward population aging; this has led to a social pattern dominated by the elderly and a smaller youth cohort in rural areas. Conversely, an increase in the birth rate would help enhance the demographic structure of these communities. As illustrated in Fig. 10(A), assuming the birth rate increases to 3% under a policy encouraging childbirth and considering migration probabilities to villages with industrial advantages, the predicted population shows an increase in the proportion of teenagers. In the long term, as the adolescent population matures and joins the labor force, the proportion of working-age

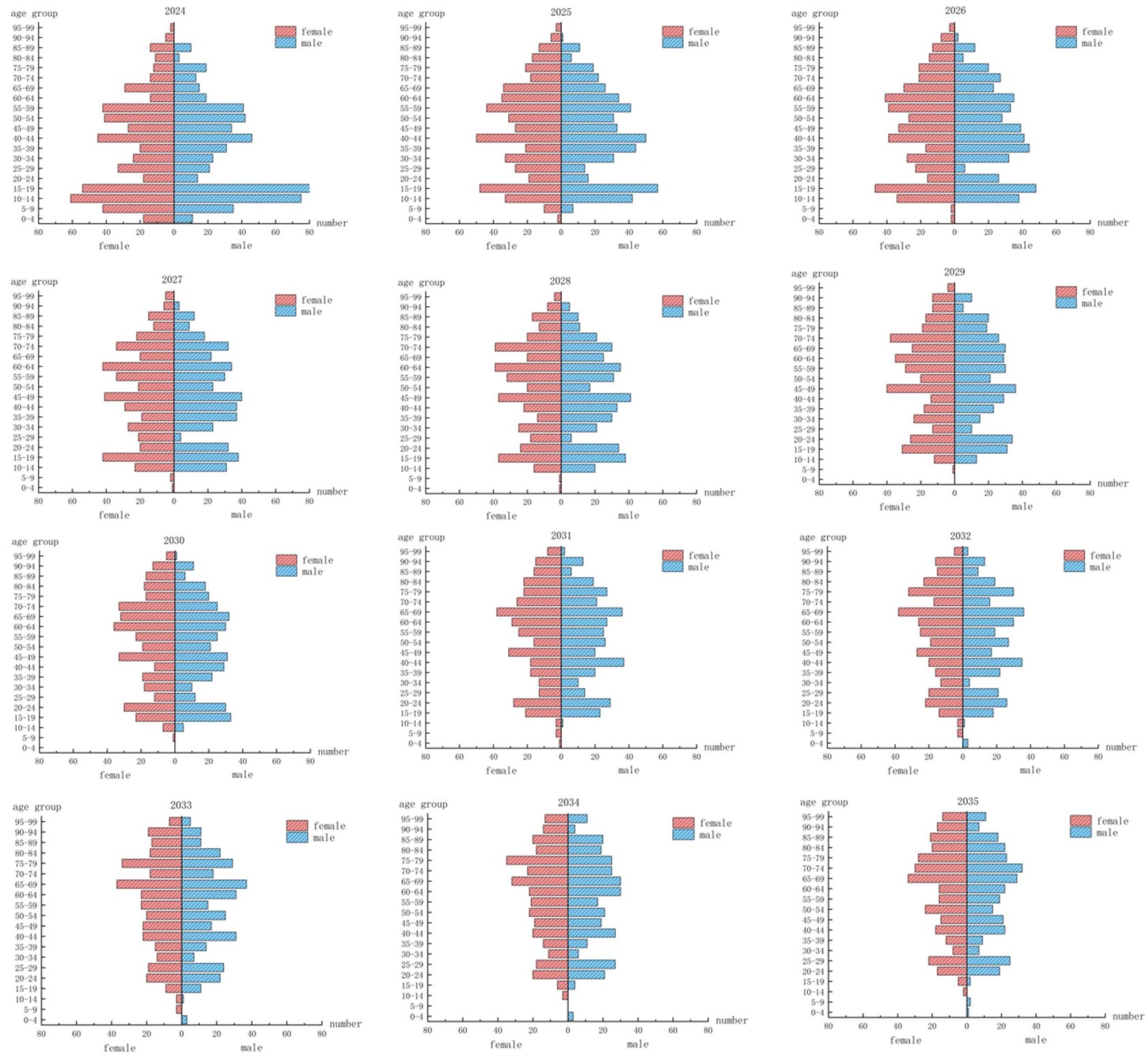

**Fig 8. Age structure of the population in Yunfeng Village with a low migration rate over the next 11 years.**

individuals is expected to rebound by 2035, providing adequate human resources for the future development of the rural economy.

**The impact of migration rates on rural population changes is significant.** The considerable loss of young and middle-aged laborers has substantially reduced the youth population in rural areas, further exacerbating the population

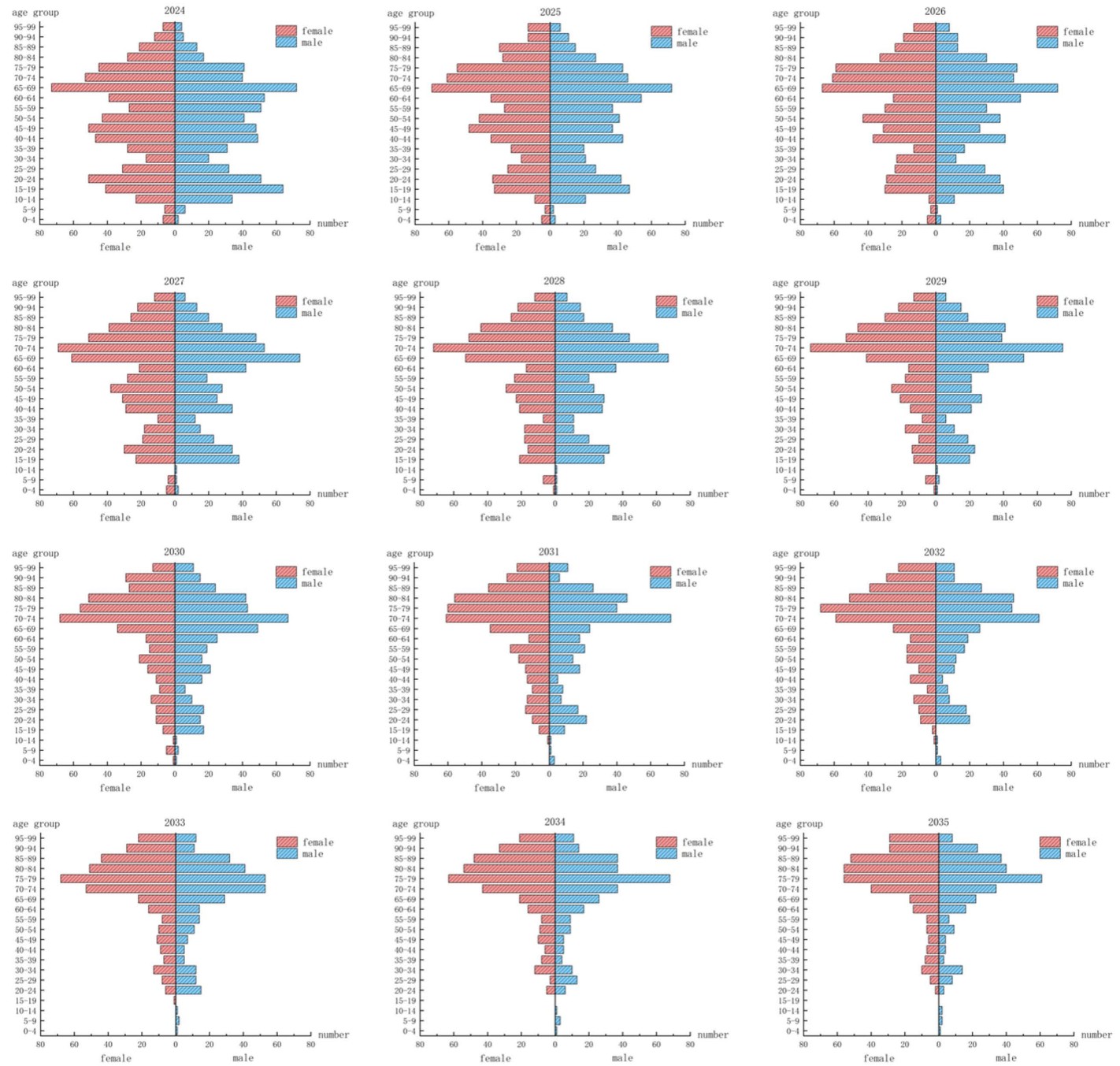

**Fig 9. Population age structure of Shihan Village with a high migration rate over the next 11 years.**

aging trend. Simulation analysis and field investigations indicate that the migrant population predominantly falls within the age range of 15–50 years, representing the primary rural labor force and adolescents receiving compulsory education. In recent years, the withdrawal of rural primary schools and decreased educational investments have caused many families

to relocate to urban areas for work and to improve their children's education. This migration pattern has intensified the outflow of young and middle-aged individuals, leaving the elderly behind.

Reducing the rate of rural population migration is crucial for mitigating the "hollowing out" effect in these communities. Fig 10 shows a simulation of a dual approach that combines high fertility rates in rural areas with a decrease in population emigration; high birth rates (3%), low migration rates (7%), and high migration rates (14%) are analyzed. Fig 10A illustrates the predicted changes in the village population under high birth and low migration scenarios (Yunfeng Village), while Fig. 10B shows the projections under high birth and high migration conditions (Shihan Village). The comparison reveals that the growth of the elderly population in Yunfeng Village has slowed. These results indicate that while encouraging birth rate is essential, rural development must also address the long-term effects of population migration. Therefore, measures should be implemented to attract talent and capital to rural areas as part of future revitalization initiatives. In addition to drawing young people back and invigorating the rural economy, promoting agricultural modernization is necessary to improve production efficiency and increase added value.

The birth rate is a crucial factor influencing the structure of rural populations. A low birth rate leads to a consistent decrease in the proportion of adolescents, further exacerbating the labor force's contraction. Additionally, the migration rate impacts rural population dynamics; particularly, the loss of young laborers intensifies the "hollowing out" trends and population aging in these areas. The simulation results of this study indicate that a combination of low migration and high birth rates would help improve the rural population structure. In light of this, four policy suggestions are proposed:

**Implementation of rural birth support policies.** Establishing local birth subsidies and improving rural social security can encourage young families to have more children. Optimizing birth support services and enhancing rural healthcare will gradually increase the proportion of young children.

**Strengthening rural education.** To ensure educational equity, measures should be taken to slow the closure of rural primary schools, increase investments in teaching staff at rural institutions, and promote the "school near" policy. This would allow rural children to receive primary and secondary education of the same quality as urban areas, mitigating population outflows caused by educational factors.

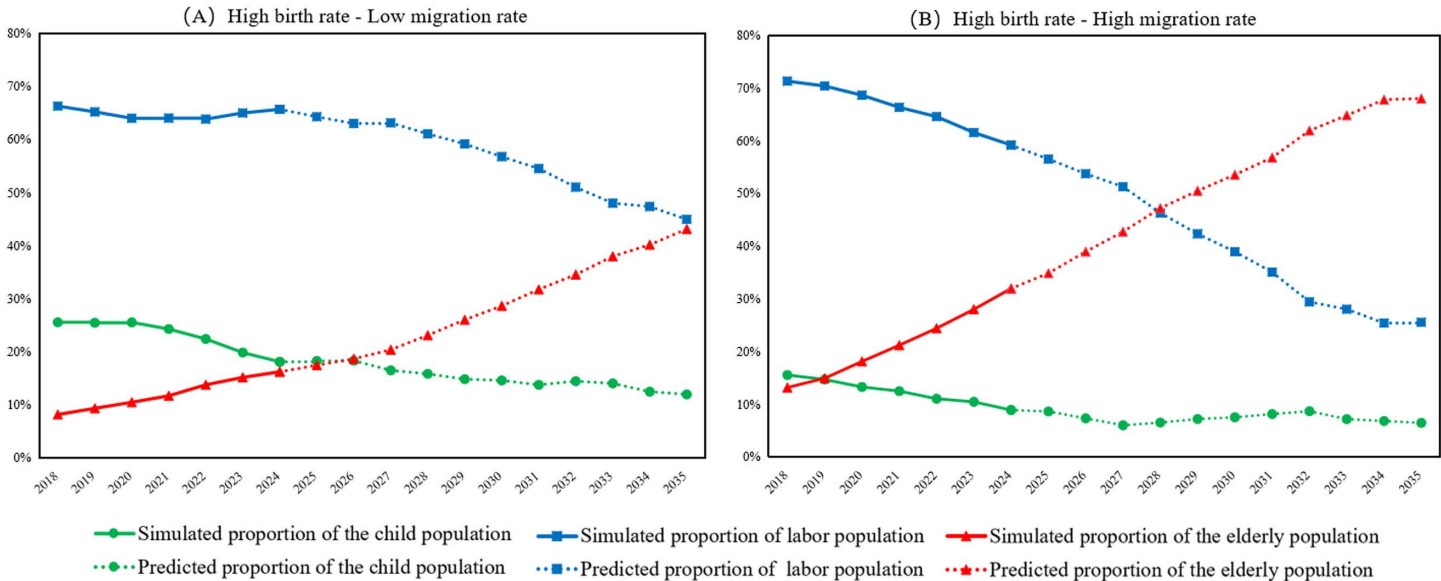

**Fig 10. Predicted influence of different birth and migration rates on rural age structure.**

**Promoting economic development.** It is essential to develop local industries to attract the return of talent people. Improving agricultural production efficiency through optimized land utilization and promoting land circulation can create more employment opportunities. Enhancing infrastructure and public services will increase household incomes, encouraging young migrant workers to return or stay in rural communities for development; this will ultimately slow down population aging and mitigate the effects of "hollowing out".

**Enhance the level of old-age security.** By raising rural pension standards, constructing a multi-tiered old-age insurance system, and strengthening the development of elderly care service facilities, we actively address the challenges posed by population aging.

## Conclusions

1. **The agent-based model can accurately simulate population dynamics:** The method effectively simulates the three major behaviors—birth, death, and migration—of individual populations through a bottom-up approach. It accurately captures changes in population size, age, and gender structure. Comparison with research data and established models confirms the robustness and reliability of the simulation outcomes.

2. **Rural population is predicted to decline continually:** Simulations based on current fertility and mortality rates indicate that total population, youth proportion, and labor force in the two representative villages will gradually decline by 2025 and continue decreasing through 2035, with youth ratios dropping below 10%. The extent of decline varies by industrial base: villages with stronger industrial support retain more young workers and experience slower decline. In contrast, villages without industrial advantages face a more pronounced population decline and an accelerated aging process. This phenomenon highlights the critical role of industrial foundations in stabilizing rural population structures.

3. **Birth and migration rates are crucial for statistically analyzing future population development in rural areas:** A lower migration rate helps maintain a relatively stable population structure in the context of a high birth rate. In contrast, a higher migration rate accelerates population aging and seriously challenges the rural labor force. These results suggest that rural revitalization should prioritize reducing migration pressure to maintain population balance and sustain the rural labor force.

## Limitations and prospects

One of the key goals of the rural revitalization strategy is to stabilize population size and optimize demographic structure. Core demographic processes such as birth, death, and migration have long-term and profound impacts on rural population dynamics. Against this backdrop, the agent-based modeling (ABM) approach offers distinct advantages in microscale simulation. It enables the dynamic modeling of individual behaviors and their interactions, thereby providing a nuanced representation of how different age groups evolve within rural settings. Compared to traditional demographic models, ABM adopts a bottom-up framework that better captures dynamic structural changes and offers more realistic and adaptable projections of future population trends. This method is particularly suitable for small-scale simulations, where localized dynamics and individual decisions play critical roles in shaping group-level outcomes. However, current ABM-based simulations still face challenges, including difficulties in data acquisition, oversimplification of behavioral rules, and the inherent complexity of large-scale agent systems. Future improvements could involve integrating richer social context data and refining behavioral assumptions to enhance model applicability, ultimately bridging the gap between micro-level simulations and macro-level demographic outcomes.

This study focuses on two types of villages in economically underdeveloped townships, and the sample size of villages is relatively small, which makes it difficult to realize a larger sample coverage. However, we have taken into account the representativeness of the village types in the sample selection, and the selected villages are typical in terms of geographic

location, industrial base and demographic structure, which can reflect the demographic evolution of the region under different development modes. Nonetheless, the results of this study need to be carefully considered when generalizing to other rural areas with significant heterogeneity, and future studies will expand the sample when conditions permit, in order to enhance the generalizability and robustness of the study's findings.

## Supporting information

**S1 Data. The minimal dataset of resident population and simulation results of two villages from 2018 to 2035.** (ZIP)

## Acknowledgments

We would like to thank reviewers for constructive comments and valuable suggestions on our manuscript.

## Author contributions

**Conceptualization:** Shitai Bao, Jianfang Wang, Siying Chen.

**Data curation:** Shanshan Huang, Yao Huang, Shitai Bao.

**Formal analysis:** Shitai Bao, Jianfang Wang.

**Investigation:** Shanshan Huang, Yao Huang, Jianfang Wang.

**Methodology:** Shitai Bao.

**Project administration:** Shitai Bao.

**Resources:** Shitai Bao.

**Software:** Shanshan Huang, Yao Huang.

**Supervision:** Shitai Bao.

**Validation:** Shanshan Huang, Yao Huang, Jianfang Wang, Siying Chen.

**Visualization:** Shanshan Huang.

**Writing – original draft:** Shanshan Huang, Yao Huang.

**Writing – review & editing:** Shitai Bao.

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
