## [Decision Letter · Decision Letter 0]

Dear Dr. Bao,

Thank you for submitting your manuscript to PLOS ONE. After careful consideration, we feel that it has merit but does not fully meet PLOS ONE’s publication criteria as it currently stands. Therefore, we invite you to submit a revised version of the manuscript that addresses the points raised during the review process.

We look forward to receiving your revised manuscript.

Kind regards,

Muhammad Umer Arshad

Academic Editor

PLOS ONE

2. For studies involving third-party data, we encourage authors to share any data specific to their analyses that they can legally distribute. PLOS recognizes, however, that authors may be using third-party data they do not have the rights to share. When third-party data cannot be publicly shared, authors must provide all information necessary for interested researchers to apply to gain access to the data. (https://journals.plos.org/plosone/s/data-availability#loc-acceptable-data-access-restrictions)

4. We note that Figures 3 and 6 in your submission contain [map/satellite] images which may be copyrighted. All PLOS content is published under the Creative Commons Attribution License (CC BY 4.0), which means that the manuscript, images, and Supporting Information files will be freely available online, and any third party is permitted to access, download, copy, distribute, and use these materials in any way, even commercially, with proper attribution. For these reasons, we cannot publish previously copyrighted maps or satellite images created using proprietary data, such as Google software (Google Maps, Street View, and Earth). For more information, see our copyright guidelines: http://journals.plos.org/plosone/s/licenses-and-copyright.

a. You may seek permission from the original copyright holder of Figures 3 and 6 to publish the content specifically under the CC BY 4.0 license. 

Additional Editor Comments:

There are some jargon and uncommon terms that make it difficult for a general reader to understand. Please revise these terms. For example, in line 34, "dwindling" can be replaced with "declining" for better clarity.

Many technical terms are used without clear definitions (e.g., "agent rules" are vaguely described).

I also noticed frequent awkward sentence structures, and some sentences are unnecessarily complex.

I suggest using active voice to improve readability.

Research methodology should be clearly defined in a separate section, including detailed information on data collection, methods, and analytical techniques.

Table 1 contains sources, but it does not add much value to the study and can be removed or revised.

The study lacks critical evaluation.

Most statements rely on secondary data without cross-verification, reducing credibility. The paper jumps between topics without logical flow, and the introduction is too broad, including historical background that does not directly contribute to the research objectives.

The choice of Agent-Based Modeling (ABM) is not justified.

The introduction fails to critically compare ABM with other micro-level modeling approaches.

There is also no discussion on why ABM is superior to alternative methods.

The study uses the Malthusian model and the Leslie model, both of which are outdated for modern demographic projections.

No sensitivity analysis has been included, making the findings less reliable. The study should also include confidence intervals or robustness checks for RMSE calculations.

The findings are not generalizable as the study is based on only two village case studies.

Additionally, the discussion section repeats points already mentioned in the results section, making it redundant. There is too much repetition throughout the paper, especially in the results and discussion sections.

The abstract and conclusion sections are weak and need to be revised to clearly highlight the study’s contributions and future research directions.

Most of the references are based on studies from China, which limits the paper’s global relevance.

Reviewers' comments:

Reviewer's Responses to Questions

**Comments to the Author**

1. Is the manuscript technically sound, and do the data support the conclusions?

Reviewer #1: Yes

Reviewer #2: Yes

Reviewer #3: Yes

2. Has the statistical analysis been performed appropriately and rigorously?

Reviewer #1: Yes

Reviewer #2: Yes

Reviewer #3: Yes

3. Have the authors made all data underlying the findings in their manuscript fully available?

Reviewer #1: No

Reviewer #2: Yes

Reviewer #3: Yes

4. Is the manuscript presented in an intelligible fashion and written in standard English?

Reviewer #1: Yes

Reviewer #2: Yes

Reviewer #3: Yes

Reviewer #1: In this study, the authors used an agent-based method to define the attributes and behavioral rules of individual agents within a population. This approach enhances predictive capability, which is particularly valuable for policymakers seeking to address the challenges of rural population decline. The paper, in general,is well written and contains valuable material. However, several sections, including the abstract, are lengthy and need shortening.

Reviewer #2: Thank you for writing up this manuscript on simulation and prediction of population using agent based modelling. Author discusses the caveats of traditional population models and recommends agent based modelling which is bottom up approach and based on 3 core behaviors – birth, death and migration. Author compares the results of agent based model with traditional methods and actual population and identifies agentic model output is more closed to the actual population than the other 2 models. Author explains the impact of birth, death, migration rates on the simulated output findings and discusses their impact and recommends policy suggestions how public officials can improve the birth rate and economy in rural areas.

Please find below comments:

1) Page 13: The population statistics mentioned in line 263 and 270 are for which year?

2) Page 15: Fig 4: Line 291: Figure represents Shihan village also has wider base, particularly age 10-14, line

291: fewer working age group 40, but this count is still bigger than Yunfeng and bigger than elderly group in

Shihan

3) Page 20: Fig 9: A & B – This figure was not discussed in the paper and predicted population of 1061 and 1324

for Yunfeng & Shihan year 2024 is not matching in this figure

4) Page 22: “Assuming birth rate increases to 3% under a policy encouraging childbirth” – during simulation birth

rate is set as 7.66%. What is the reason to mention birth rate as 3% ?

Reviewer #3: 1. Strengthen the Discussion of Novelty in Comparison to Past Models

Clearly state how ABM differs from and improves upon traditional models.

2. Justify Parameter Selection More Rigorously

Explain why birth rates, migration probabilities, and mortality rates were chosen.

3. Improve Conciseness and Clarity in the Results Discussion

Avoid repeating numbers already presented in tables; instead, summarize trends.

Improve readability with subheadings and bold key findings.

4. Expand Policy Implications and Real-World Applications

Provide specific policy recommendations, not just general suggestions. Discuss how the findings can be applied to different rural contexts beyond China.

**Do you want your identity to be public for this peer review?** For information about this choice, including consent withdrawal, please see our Privacy Policy

Reviewer #1: No

Reviewer #2: No

Reviewer #3: **Yes: ** Nora

---

## [Author Response · Author response to Decision Letter 1]

24 Mar 2025

We sincerely thank the reviewers for their constructive comments and valuable suggestions. Our responses to each point are provided below

Editorial comments

1.There are some jargon and uncommon terms that make it difficult for a general reader to understand. Please revise these terms. For example, in line 34, "dwindling" can be replaced with "declining" for better clarity. Many technical terms are used without clear definitions (e.g., "agent rules" are vaguely described). I also noticed frequent awkward sentence structures, and some sentences are unnecessarily complex. I suggest using active voice to improve readability.

Thank you for your thorough review and valuable feedback. Concerning the issues of terminology usage and sentence structure that impact readability, we have conducted a comprehensive review of the technical terms and refined the sentence expressions throughout the manuscript. Additionally, we have sought professional guidance to ensure clarity and precision. We believe this revised version offers a more intuitive and accessible presentation.

2. Research methodology should be clearly defined in a separate section, including detailed information on data collection, methods, and analytical techniques.Table 1 contains sources, but it does not add much value to the study and can be removed or revised.

(1) This study's research methodology, which includes the agent-based simulation approach, the constructed agent behavior rules, and population validation methods, has been presented in a separate section.

(2) Table 1 presents the behavioral rules and probabilistic parameters for population agents across various age groups, derived from empirical data on birth rates, mortality rates, and migration rates obtained through rural surveys. These elements serve as a critical foundation for simulating individual agent behaviors. To ensure scientific rigor and transparency in the simulation process, we have retained this table while enhancing its content for better clarity and readability.

3.The choice of Agent-Based Modeling (ABM) is not justified. The introduction fails to critically compare ABM with other micro-level modeling approaches.There is also no discussion on why ABM is superior to alternative methods.

Concerning the insufficient rationale for choosing agent-based modeling (ABM), we have added a literature review in the introduction comparing ABM with other static simulation methods and providing additional justification for selecting this approach.

4.The study uses the Malthusian model and the Leslie model, both of which are outdated for modern demographic projections. No sensitivity analysis has been included, making the findings less reliable. The study should also include confidence intervals or robustness checks for RMSE calculations.

(1) The Malthusian model, which only predicts total population size, is somewhat outdated. In the revised version, we have removed it from the validation process. The Leslie model, a classic demographic model capable of predicting both population size and structure, is primarily used in this study to compare ABM simulation results across different age groups, thereby enhancing the cross-validation of simulation results.

(2) The key factors in this study’s simulation are birth, mortality, and migration rates. While the results section does not include a formal sensitivity analysis, the discussion section already explores simulations under high/low birth rates and high/low migration rates, effectively reflecting aspects of sensitivity analysis. In the revised manuscript, we have renamed Section 4.3 from “Discussion on Rural Population Change” to “Sensitivity Analysis and Policy Recommendations” to highlight this aspect better.

(3) Confidence Interval: Traditionally, RMSE is a point estimate that directly reflects the average prediction error of the entire dataset and is not typically associated with a confidence interval. In the revised version, the paired-sample T-test was used to verify the difference of the simulated and actual values of the two-village population using SPSS 27. Furthermore, Bootstrap for Paired Sampled Test was checked. The results sufficiently demonstrate that the simulated values closely match the actual values (difference equaling zero at 95% confidence level). For details, see Tables 3 and 4.

5.The findings are not generalizable as the study is based on only two village case studies. Additionally, the discussion section repeats points already mentioned in the results section, making it redundant. There is too much repetition throughout the paper, especially in the results and discussion sections.

(1) Although the study area consists of two village case studies, they represent two distinct types of villages—one with industrial advantages and another with industrial disadvantages. Our objective is to explore the changing patterns of the rural population using these two opposing cases, which can be extended to similar villages. In the revised manuscript, the representativeness of each village has been explicitly enhanced, and the findings from these specific cases have been applied to a broader context.

(2) Regarding content redundancy, we have streamlined the results section to avoid duplicating the same data and findings in the discussion. Instead, we focused on interpreting the underlying reasons for these results, their connections to existing studies, and their implications for future research and practice. This revision also enhances the coherence and readability of the paper.

6.The abstract and conclusion sections are weak and need to be revised to clearly highlight the study’s contributions and future research directions.

(1) Abstract: Three key points have been clarified further: (i) the differences between agent-based modeling (ABM) and traditional demographic models, emphasizing ABM’s dynamic simulation capabilities and bottom-up approach; (ii) a clear comparative explanation of population changes between the two types of villages; and (iii) an interpretation of key parameters’ sensitivity, highlighting migration and birth rates as the main drivers of rural population change. Finally, we strengthened the policy relevance by linking the research findings directly to policy recommendations, emphasizing their practical implications.

(2) Conclusion: The discussion of results has been streamlined to avoid redundancy with numerical data already presented in tables, while summarizing the population change trends in the two types of villages. The study’s contributions are explicitly highlighted, expanding on its policy implications and practical applications. Specific policy recommendations and future research directions have been provided, reinforcing the applicability of the findings to other rural areas and assessing their generalizability. Additionally, the discussion section further emphasizes the strengths and limitations of ABM in population simulation, outlining potential avenues for future methodological improvements.

7.Most of the references are based on studies from China, which limits the paper’s global relevance.

Thank you for your suggestion regarding the scope of references. To enhance the model methodology's global applicability, we have incorporated several international studies.

Reviewer's Comments

Reviewer #1: In this study, the authors used an agent-based method to define the attributes and behavioral rules of individual agents within a population. This approach enhances predictive capability, which is particularly valuable for policymakers seeking to address the challenges of rural population decline. The paper, in general,is well written and contains valuable material. However, several sections, including the abstract, are lengthy and need shortening.

Thank you for your recognition. The abstract has been refined to focus on the core contributions (ABM's predictive capability and policy recommendations), with redundant background descriptions removed. Other sections, such as the literature review, have been condensed to essential content. At the same time, the conclusion and discussion have been further streamlined to ensure a concise and coherent structure throughout the paper.

Reviewer #2: Thank you for writing up this manuscript on simulation and prediction of population using agent based modelling. Author discusses the caveats of traditional population models and recommends agent based modelling which is bottom up approach and based on 3 core behaviors – birth, death and migration. Author compares the results of agent based model with traditional methods and actual population and identifies agentic model output is more closed to the actual population than the other 2 models. Author explains the impact of birth, death, migration rates on the simulated output findings and discusses their impact and recommends policy suggestions how public officials can improve the birth rate and economy in rural areas.

Please find below comments:

1) Page 13: The population statistics mentioned in line 263 and 270 are for which year?

2) Page 15: Fig 4: Line 291: Figure represents Shihan village also has wider base, particularly age 10-14, line291: fewer working age group 40, but this count is still bigger than Yunfeng and bigger than elderly group in Shihan

3) Page 20: Fig 9: A & B – This figure was not discussed in the paper and predicted population of 1061 and 1324 for Yunfeng & Shihan year 2024 is not matching in this figure

4) Page 22: “Assuming birth rate increases to 3% under a policy encouraging childbirth” – during simulation birth rate is set as 7.66%. What is the reason to mention birth rate as 3% ?

(1) On page 13, the demographic data mentioned in lines 263 and 270 were obtained from the 2024 field survey.

(2) In Figure 3 (line 291), the description has been further enhanced, and it has been verified that the figure represents the age structure of the resident population in two villages based on actual survey data.

(3) On page 20, additional descriptions have been provided for parts A and B of Figure 8, explaining the simulated population changes and demographic structure changes. The data in the figure have also been verified, with detailed modifications in Figure 8.

(4) On page 22, the initial birth rate in the simulation was 7.66‰, which was later increased to 3% under the assumption of an unrestricted pro-birth policy. This adjustment aimed to explore village population changes and validate the hypothesis that "birth rate is a key factor influencing rural demographic structure," providing a basis for subsequent policy recommendations.

Reviewer #3: 1. Strengthen the Discussion of Novelty in Comparison to Past Models

Clearly state how ABM differs from and improves upon traditional models.

2. Justify Parameter Selection More Rigorously Explain why birth rates, migration probabilities, and mortality rates were chosen.

3. Improve Conciseness and Clarity in the Results Discussion Avoid repeating numbers already presented in tables; instead, summarize trends. Improve readability with subheadings and bold key findings.

4. Expand Policy Implications and Real-World Applications. Provide specific policy recommendations, not just general suggestions. Discuss how the findings can be applied to different rural contexts beyond China.

Thank you for your review. Below are my responses to your suggestions and the revisions made to the manuscript.

(1) Regarding the discussion on enhancing the novelty compared to past models, a literature review was added to the introduction section, comparing Agent-Based Modeling (ABM) with other static simulation methods. This highlights the distinctions and improvements of ABM over traditional models and explains the rationale for choosing the ABM approach. Second, the discussion was revised to emphasize the differences between Agent-Based Models (ABM) and traditional models. The advantages of ABM in capturing individual-level heterogeneity and dynamic interactions were underscored, aspects that are often oversimplified in traditional methods.

(2) Regarding stricter justification for parameter selection: The revised manuscript provides more detailed justification for selecting parameters such as birth rate, migration probability, and mortality rate. The empirical and theoretical foundations for these parameters are explained based on existing literature and relevant statistical data. Additionally, robustness tests for the model were added and included in the "Verification of simulation results" section.

(3) Improving the conciseness and clarity of the results discussion: The results section was modified to enhance clarity and conciseness in the revised manuscript. The focus was placed on summarizing key trends and policy recommendations, with the policy recommendations section bolded to improve readability and facilitate better navigation of the results. These changes are reflected in the "Sensitivity analysis and policy implications" and "Conclusions" sections.

(4) Expanding policy implications and practical applications: The revised manuscript expands on policy implications and proposes specific, actionable policy recommendations based on the research findings. Regarding population birth, it suggests lifting restrictions on rural population fertility and increasing childbirth and childcare subsidies. To address labor migration loss, it proposes supporting agricultural industries, increasing farming subsidies, enhancing land consolidation and transfer, and boosting agricultural income to attract population return and reduce outmigration. For educational migration, it recommends retaining village primary schools, slowing down school closures, increasing care for left-behind children, and investing in rural education to reduce population outflow due to education. These details can be found in the "Sensitivity analysis and policy implications" section.

---

## [Decision Letter · Decision Letter 1]

Dear Dr. Bao,

Thank you for submitting your manuscript to PLOS ONE. After careful consideration, we feel that it has merit but does not fully meet PLOS ONE’s publication criteria as it currently stands. Therefore, we invite you to submit a revised version of the manuscript that addresses the points raised during the review process.

We look forward to receiving your revised manuscript.

Kind regards,

Muhammad Umer Arshad

Academic Editor

PLOS ONE

Journal Requirements:

Additional Editor Comments:

Thank you for your revised submission. The Author made substantial improvements, but several issues need to be addressed.

The heading Experimental area, it is not appropriate for a simulation-based study; Please change it to Study area or Case study description.

The study is based on two villages, which limits the broader applicability. Please acknowledge this limitation in the discussion and limitations section, the same comment has been not addressed in revision

Several sentences still remain verbose and repetitive; the same comment was given in the last revision, but has not been appropriately addressed.

Please revise the language carefully and remove all unnecessary wording. Cut down each and make it clear and short.

Figures 9 to 12 lack sufficient explanation. Please ensure that each figure is discussed in detail and its relevance to the research question.

Regarding the sensitivity analysis comment, rather than doing the sensitivity analysis, the authors just revised the section name “Discussion about rural population change” and labeled it as a sensitivity analysis, but does this meet the expected standards? Please justify.

A proper sensitivity analysis should include variation of model parameters across a plausible range with cross-splicing output evaluation. I believe the author should revise the name of this section, as this does not represent the as sensitivity analysis

The conclusion section is too long, please write a concise conclusion in 1-2 paragraphs.

Reviewers' comments:

Reviewer's Responses to Questions

**Comments to the Author**

Reviewer #4: (No Response)

2. Is the manuscript technically sound, and do the data support the conclusions?

Reviewer #4: Yes

3. Has the statistical analysis been performed appropriately and rigorously?

Reviewer #4: Yes

4. Have the authors made all data underlying the findings in their manuscript fully available?

Reviewer #4: No

5. Is the manuscript presented in an intelligible fashion and written in standard English?

Reviewer #4: Yes

Reviewer #4: The authors conducted a simulation and prediction of population using an agent based modelling on 3 factors: birth, death and migration. Overall, the manuscript is clearly written and presents a novel approach to the topic. However, several revisions are needed to improve the clarity and coherence of the presentation.

1. based on your response to reviewers, the validation part of Malthusian model is excluded in the latest manuscript, but in the abstract, it’s still showing ‘accuracy validated through Malthusian model’ (also see comment #6)

2. line 110-111, ‘(2) these models typically, these models concentrate on macro-level predictions…’ has a problem in sentence structure

3. line 120, the authors introduce intelligent agent approach rather abruptly. Although the authors mention a simulation approach in the previous paragraph, it’s better to add a clear transition to improve the flow.

4. line 226-229, xi(t), bi, di can be typed using equation editor for clearer notations and properly aligned subscripts, like the authors do after equation 2

5. the first paragraph of Population simulation section is a little confusing and contradictory. It seems like Xiache town is considered to be a relatively underdeveloped area (Guangdong for sure cannot be called the underdeveloped area), but it is also the provincial center and characterized by a high degree of specialization ad high incomes. Geographically, Hepin county is located in the northeast part of Guangdong, and not really part of the pearl river delta. ‘the town serves as the provincial center’ - Xiache town is neither the capital of the Guangdong province nor able to serve as the center of province, given its limited resources.

6. tables 3-5 Malthus model results are NA, but in line 300-301, the authors mention ‘results were subsequently verified against the predictions generated by Malthus model’. If the results are not included and showed as NA only, the columns should be removed.

7. tables 3-4, please add the unit for (Malthus model,) Leslile model, agent simulation, current population

8. units: the authors sometimes use km, ha, %, ‰, as units. It would be better to keep the units consistent. Similarly, line 392, ‘the proportion of individuals aged 15–19 will continue to decrease, while the percentage of the population over 70 years will steadily increase.’ The sentence sounds a bit awkward. While proportion and percentage can be used interchangeably, they also refer to slightly different concepts.

9. As the editor previously commented, the findings might not be generalizable given the study is based on two villages. The authors interpret the two villages as one with industrial advantages and the other without, both located in a relatively underdeveloped town. Using industrial (or economic) status alone might oversimplify the rural Chinese demographic diversity. Rural doesn’t necessarily imply underdeveloped, especially those close to major cities. Development level, accessibility, demographic profiles in rural China can be very different. Social, cultural, policy-related, educational and geographic factors can also contribute to rural population change. Nowadays some young people choose to migrate to rural areas because they want slower-paced lifestyle or lower living costs. Conversely, some people remain in their relatively poor remote villages because of strong family networks, ethnic identity or attachment to ancestral land. If a rural town's economy relies primarily on historical preservation, or national park tourism, will the town be classified as underdeveloped or industrially disadvantaged?

10. what are the study limitations?

**Do you want your identity to be public for this peer review?** For information about this choice, including consent withdrawal, please see our Privacy Policy

Reviewer #4: No

---

## [Author Response · Author response to Decision Letter 2]

24 Apr 2025

We sincerely appreciate the constructive comments and valuable suggestions provided by the reviewer and editor. Our detailed responses to each point are below.

Additional Editor Comments:

1. The heading Experimental area, it is not appropriate for a simulation-based study; Please change it to Study area or Case study description.

The original use of the term“Experimental area”did not accurately reflect the nature of the simulation study. It has now been revised to“Overview of the study area”in the relevant heading, as reflected in the revised manuscript (see line 186).

2. The study is based on two villages, which limits the broader applicability. Please acknowledge this limitation in the discussion and limitations section, the same comment has been not addressed in revision

We agree with the possible limitations of the simulation analysis based only on two villages. However, this study focuses on two types of villages in economically underdeveloped towns and townships, respectively reflecting the two types of villages with local industrial advantages and disadvantages, as well as their population change scenarios. Perhaps the number of selected village samples is limited, but the two types of villages still have obvious representativeness in terms of industrial differences and population changes. They can reflect the characteristics of rural population evolution under industrial differences, which also exist widely in other rural areas in China.

The sample selection of this study may not reflect various types of villages and the multi-dimensional causes influencing population changes. The section of "Limitations and prospects" was added in the latest revised manuscript. However, we mainly classified the two types of villages from the perspective of rural industrial differences and deeply explored the population evolution of the two types of villages triggered thereby. The focus of the dimension analyzed in this study lies in the impact of industrial differences on the population, which is somewhat universal in rural areas across the country.

We believe that the constructed simulation framework and method are universal in villages with obvious industrial differences and their population evolution, and can provide theoretical references and modeling paradigms for the population evolution and prediction of other villages with similar industrial economic differences.

3. Several sentences still remain verbose and repetitive; the same comment was given in the last revision, but has not been appropriately addressed. Please revise the language carefully and remove all unnecessary wording. Cut down each and make it clear and short.

We have carefully reviewed the entire manuscript, with a focus on checking and streamlining lengthy, repetitive, or unclear sentences. In particular, we have comprehensively optimized the language in the sections of theoretical foundation, model design, and analysis & discussion by addressing redundant wording. This revision removed unnecessary conjunctions, repetitive expressions, and explanatory sentences to make the sentence structure more compact and concise.

4. Figures 9 to 12 lack sufficient explanation. Please ensure that each figure is discussed in detail and its relevance to the research question.

There were only 10 figures in the previous revised the manuscript, rather than 12. To address the issue of insufficient explanations, we have provided further supplementary clarifications for the corresponding figures in accordance with the suggestions, ensuring that the relationship between the content, variation trends, and research questions of each figure is explicitly explained. Specific details can be found in the red-marked text of the revised manuscript.

5. Regarding the sensitivity analysis comment, rather than doing the sensitivity analysis, the authors just revised the section name “Discussion about rural population change” and labeled it as a sensitivity analysis, but does this meet the expected standards? Please justify. A proper sensitivity analysis should include variation of model parameters across a plausible range with cross-splicing output evaluation. I believe the author should revise the name of this section, as this does not represent the as sensitivity analysis

In the previous revised manuscript, we only adjusted the title of the section "Discussion on rural population change," which stemmed from our understanding deviation for the concept of "sensitivity analysis". In fact, the core parameters of this simulation study are the birth rate and migration rate. In the latest revised manuscript, a section titled " The impact of births, deaths and migration rates on population change is updated. We adjusted the above-mentioned parameters within a reasonable range and systematically analyzed their effects on the population structure and change trends in the simulation results. From the perspective of the analytical purpose, this essentially aims to evaluate the responsiveness of the model output to changes in key input parameters, which is consistent with the core objectives of sensitivity analysis. Therefore, this section actually reflects the model's response characteristics to fluctuations in core parameters and possesses the essential attributes of sensitivity analysis. We consider it logically reasonable and well-founded to regard this as a sensitivity analysis of the model parameters.

6. The conclusion section is too long, please write a concise conclusion in 1-2 paragraphs.

We have refined the original conclusions with the core content while striving for concise language and a compact structure. To enhance the sense of organization and readability, we have maintained the form of point-by-point summarization, in which the first sentence of each point provides a more succinct generalization of the corresponding research conclusion.

Reviewers' comments:

The authors conducted a simulation and prediction of population using an agent based modelling on 3 factors: birth, death and migration. Overall, the manuscript is clearly written and presents a novel approach to the topic. However, several revisions are needed to improve the clarity and coherence of the presentation.

1. based on your response to reviewers, the validation part of Malthusian model is excluded in the latest manuscript, but in the abstract, it’s still showing ‘accuracy validated through Malthusian model’ (also see comment #6)

We would like to express our gratitude to the reviewers for carefully examining the details of the manuscript. In accordance with your suggestions, we have removed the expression " accuracy validated through Malthusian model " from the abstract of the first revised version. Additionally, we have made corresponding adjustments to the content of the abstract to ensure its consistency with the main body of the latest revised manuscript.

2. line 110-111, ‘(2) these models typically, these models concentrate on macro-level predictions…’ has a problem in sentence structure

In the second revised version of the manuscript, we have made adjustments to the grammatical structure of this sentence to enhance the accuracy and readability. The revised sentence is updated in line 70

3. line 120, the authors introduce intelligent agent approach rather abruptly. Although the authors mention a simulation approach in the previous paragraph, it’s better to add a clear transition to improve the flow.

We have carefully examined the structure of this paragraph and recognized that when introducing the agent-based modeling (ABM) approach, there is indeed a lack of necessary logical transition. To enhance the coherence between paragraphs, before introducing the ABM model, we have added a transitional sentence in line 73. This clearly points out that the deficiencies of traditional models have prompted researchers to turn to more flexible methods with the ability to represent individuals, thus naturally leading to the advantages of the agent-based model.

4. line 226-229, xi(t), bi, di can be typed using equation editor for clearer notations and properly aligned subscripts, like the authors do after equation 2

In accordance with your suggestions, we have re-entered the variables such as xi(t), bi, and di in lines 226–229 using a formula editor. This ensures that the subscripts and variable symbols are clear and standardized, and the typesetting is consistent. Meanwhile, we have also conducted a thorough check of the entire manuscript to ensure that the expressions of other similar mathematical symbols conform to the typesetting specifications.

5. the first paragraph of Population simulation section is a little confusing and contradictory. It seems like Xiache town is considered to be a relatively underdeveloped area (Guangdong for sure cannot be called the underdeveloped area), but it is also the provincial center and characterized by a high degree of specialization ad high incomes. Geographically, Heping county is located in the northeast part of Guangdong, and not really part of the pearl river delta. ‘the town serves as the provincial center’ - Xiache town is neither the capital of the Guangdong province nor able to serve as the center of province, given its limited resources.

We have rephrased the content of this paragraph, clearly pointing out that Xiache Town is located in Heping County, in the northeastern part of Guangdong Province. It is a typical township characterized by relatively backward economy and significant population outflow. We have deleted the ambiguous expression such as "provincial central town" and avoided the statement that equates the whole of Guangdong Province, so as to ensure that the content description is consistent with the actual geographical location and economic characteristics. In the article, Heping County is located in Heyuan City, and Heyuan City is an outer city of the Pearl River Delta. The intention of the expression in the article is not to claim that Xiache Town is the provincial capital or the provincial center, See line 193 for details.

6. tables 3-5 Malthus model results are NA, but in line 300-301, the authors mention ‘results were subsequently verified against the predictions generated by Malthus model’. If the results are not included and showed as NA only, the columns should be removed.

We would like to express our sincere gratitude for your valuable suggestions, esteemed expert. It is possible that the first revised version of the manuscript confused the reviewer. In the second-round revised version, all the relevant results and descriptions pertaining to the Malthusian model have been completely removed, and the related sentences have been updated accordingly.

7. tables 3-4, please add the unit for (Malthus model,) Leslile model, agent simulation, current population

We have made corresponding revisions to Table 3 and Table 4. The unit of the population in the tables has been uniformly marked as "persons". In addition, considering that the content related to the Malthusian model has been removed from the main text during the revision process, we have not retained the relevant information in the tables to avoid misunderstandings.

8. units: the authors sometimes use km, ha, %, ‰, as units. It would be better to keep the units consistent. Similarly, line 392, ‘the proportion of individuals aged 15–19 will continue to decrease, while the percentage of the population over 70 years will steadily increase.’ The sentence sounds a bit awkward. While proportion and percentage can be used interchangeably, they also refer to slightly different concepts.

We have unified the expressions of units throughout the entire text. The unit of the area of the relevant jurisdiction has been uniformly changed from km² to ha. For specific revisions, please refer to the section titled "Overview of the study area". In addition, concerning the problem in the expression of line 392, we have re-examined the logic and wording of that sentence and made revisions. We have used "proportion" as the unit for the population. For specific revisions, please refer to line 320 of the revised manuscript.

9. As the editor previously commented, the findings might not be generalizable given the study is based on two villages. The authors interpret the two villages as one with industrial advantages and the other without, both located in a relatively underdeveloped town. Using industrial (or economic) status alone might oversimplify the rural Chinese demographic diversity. Rural doesn’t necessarily imply underdeveloped, especially those close to major cities. Development level, accessibility, demographic profiles in rural China can be very different. Social, cultural, policy-related, educational and geographic factors can also contribute to rural population change. Nowadays some young people choose to migrate to rural areas because they want slower-paced lifestyle or lower living costs. Conversely, some people remain in their relatively poor remote villages because of strong family networks, ethnic identity or attachment to ancestral land. If a rural town's economy relies primarily on historical preservation, or national park tourism, will the town be classified as underdeveloped or industrially disadvantaged?

We acknowledge your concerns regarding the categorization of rural areas based on the development status of a specific industry. This approach may lead to oversimplification and may not fully capture the complex factors driving population changes in rural China. The sample selection of this study may not have reflected all kinds of rural types and the multi-dimensional causes influencing population changes.

In the study, two villages with typical characteristics but different industry levels were selected as case studies. These two villages respectively reflect different population migration scenarios caused by local industrial advantages and disadvantages, and such type differences are also widely prevalent in other similar rural areas across the country. We mainly classified two types of rural areas from the dimensions of industry and population development status, and conducted an in-depth exploration of the two different population migration situations. It is undeniable that various other factors can also lead to population changes in rural areas. Nevertheless, the focus of the analysis dimensions in this study lies in the impact of industry on the population, which has a certain degree of universality among rural areas nationwide.

In the revised manuscript, we have added a section titled "Limitations and prospects". Relevant content has been supplemented in the explanation of the research limitations, clearly pointing out that the number of village samples in the current study is limited. We also suggest that future research conduct model verification and application in a broader regional context to further enhance the adaptability and reliability of the simulation framework. For specific revisions, please refer to the relevant content after line 429.

Finally, regarding the question "If a rural town's economy relies primarily on historical preservation, or national park tourism, will the town be classified as underdeveloped or industrially disadvantaged?", my answer is negative. If the economy of a rural town relies on historical preservation or national park tourism, it is not necessarily classified as "underdeveloped" or "industrially disadvantaged". The key lies in whether its tourism industry has economies of scale, driving capabilities, stability, and the degree of integration with local social resources.

10. what are the study limitations?

Although agent-based modeling (ABM) provides an effective tool for depicting the relationship between the behaviors of microscopic individuals and macroscopic population dynamics, its application in simulating rural population changes still has certain limitations. Therefore, the limitations of the study have been supplemented in the revised manuscript, and the future research directions have been indicated. For specific revisions, please refer to the newly added section " Limitations and prospects", that is, the content after line 428.

---

## [Editor Report · Decision Letter 2]

Simulation and prediction of rural population changes using agent-based models

PONE-D-25-02914R2

Dear Dr. Bao,

We’re pleased to inform you that your manuscript has been judged scientifically suitable for publication and will be formally accepted for publication once it meets all outstanding technical requirements.

Kind regards,

Muhammad Umer Arshad

Academic Editor

PLOS ONE
---

## [Editor Report · Acceptance letter]

PONE-D-25-02914R2

PLOS ONE

Dear Dr. Bao,

I'm pleased to inform you that your manuscript has been deemed suitable for publication in PLOS ONE. Congratulations! Your manuscript is now being handed over to our production team.

Kind regards,

on behalf of

Dr. Muhammad Umer Arshad

Academic Editor

PLOS ONE